# Alterations in Corneal Morphology and Thickness Associated with Methylphenidate Treatment in Children with Attention-Deficit/Hyperactivity Disorder

**DOI:** 10.3390/diagnostics15182368

**Published:** 2025-09-18

**Authors:** Fatma Sumer, Merve Yazici

**Affiliations:** 1Department of Ophthalmology, Faculty of Medicine, Recep Tayyip Erdogan University, Rize 53100, Türkiye; 2Department of Child and Adolescent Psychiatry, Faculty of Medicine, Recep Tayyip Erdogan University, Rize 53100, Türkiye

**Keywords:** methylphenidate, Attention-Deficit/Hyperactivity Disorder (ADHD), corneal endothelium, specular microscopy, intraocular pressure

## Abstract

**Background/Objectives**: Although methylphenidate is a first-line pharmacological agent in the treatment of Attention-Deficit/Hyperactivity Disorder (ADHD), its long-term effects on ocular tissues, particularly the corneal endothelium, remain poorly understood. Given the cornea’s metabolic sensitivity, subclinical changes may occur even in the absence of overt ophthalmologic symptoms. This study aims to evaluate the impact of six-month methylphenidate treatment on corneal endothelial morphology and intraocular pressure (IOP) in pediatric patients with ADHD. **Methods**: This prospective observational study included 100 treatment-naive children with ADHD and 100 age- and sex-matched healthy controls. All participants underwent comprehensive ophthalmologic assessment at baseline. In the ADHD group, follow-up evaluations were performed after six months of methylphenidate therapy. Endothelial cell density (ECD), average cell area (AVE), standard deviation (SD), coefficient of variation (CV), hexagonality index (6A), central corneal thickness (CCT), and IOP were measured using specular microscopy and corneal topography. ADHD symptom severity was evaluated using the Turgay DSM-IV-Based Rating Scale. **Results**: Significant reductions in ECD and increases in CCT, CV, AVE, and SD were observed following treatment (*p* < 0.001). IOP also showed a statistically significant increase while remaining within normal physiological limits. Weak but significant correlations were found between inattention scores and ECD (r = 0.222), and between inattention and corneal volume (r = −0.248). **Conclusions**: Chronic methylphenidate use may be associated with measurable changes in corneal endothelial microstructure and IOP in children with ADHD. These findings highlight the need for routine ophthalmologic monitoring during stimulant therapy and underscore the importance of further large-scale, long-term studies exploring the neuro-ophthalmologic implications of pediatric psychopharmacological treatment.

## 1. Introduction

Attention-Deficit/Hyperactivity Disorder (ADHD) is a common neurodevelopmental condition characterized by persistent symptoms of inattention, hyperactivity, and impulsivity that are inconsistent with developmental norms and significantly impair academic, social, and functional performance [1]. The global prevalence of childhood ADHD is estimated to range between 5% and 7% [2]. Current treatment strategies for ADHD employ a multimodal approach that combines pharmacological and behavioral interventions. Among pharmacologic agents, short- and long-acting formulations of methylphenidate are widely accepted as first-line therapies in Europe due to their robust and well-documented efficacy in reducing core ADHD symptoms [3].

Although the central nervous system is the primary target of methylphenidate, its influence may extend to peripheral systems, including the visual system. The ocular effects of methylphenidate are thought to arise indirectly through the modulation of dopaminergic and noradrenergic pathways [4]. Dopaminergic stimulation in retinal neurons may alter photoreceptor adaptation and contrast sensitivity, while elevated norepinephrine levels may transiently affect pupillary dilation (mydriasis) and accommodation responses. These neuropharmacological effects, particularly during long-term or high-dose exposure, have been reported to influence visual function and ocular perfusion [5]. Additionally, increased sympathetic tone may induce vasoconstriction in the anterior segment vasculature, potentially eliciting subclinical stress responses in metabolically sensitive ocular tissues such as the cornea.

The corneal endothelium plays a critical role in preserving corneal transparency and structural stability [6]. Reductions in endothelial cell density, morphological irregularities, and alterations in central corneal thickness (CCT) can compromise corneal physiology and degrade visual quality [7]. Therefore, an objective evaluation of corneal morphology in pediatric patients receiving chronic methylphenidate treatment holds clinical importance.

Previous research investigating the ocular effects of methylphenidate has yielded limited and inconsistent findings. Case reports have documented visual disturbances including accommodation disorders, mydriasis, blurred vision, and dry eyes following methylphenidate treatment [8,9]. Soyer et al. reported a 9-year-old child with significant visual acuity decline secondary to accommodation disorder after methylphenidate and lisdexamfetamine therapy [9].

Small-scale prospective studies have further suggested subtle structural effects, with Larrañaga-Fragoso et al. demonstrating reduced anterior chamber depth after 9 months of treatment [10]. Isolated case reports have also described associations between methylphenidate use and intraocular pressure elevation, with some cases progressing to glaucoma and cataract formation [11].

However, despite these observations, comprehensive investigations systematically evaluating corneal endothelial morphology in pediatric populations under chronic methylphenidate therapy remain scarce. Most existing studies have been limited by small sample sizes (typically 14–22 patients), short follow-up periods, cross-sectional designs, and lack of standardized corneal assessment protocols [12,13].

This underscores a critical gap in the literature and highlights the urgent need for well-designed prospective studies with adequate sample sizes, extended longitudinal follow-up, and standardized corneal imaging protocols to definitively clarify the long-term structural effects of methylphenidate on corneal health in children and adolescents.

This study aims to objectively evaluate the potential ocular effects of long-term methylphenidate use in children and adolescents diagnosed with ADHD. Specifically, we investigated changes in corneal endothelial cell density, cellular morphology, and central corneal thickness using standardized imaging techniques. Inclusion of an age- and gender-matched healthy control group enabled direct comparison. By bridging the fields of child psychiatry and ophthalmology, this research seeks to contribute to the limited body of literature on this topic and support the integration of ophthalmologic monitoring into the routine care of stimulant-treated pediatric populations.

## 2. Materials and Methods

### 2.1. Study Design and Participants

This prospective, observational case–control study was conducted at a tertiary university hospital after receiving approval from the Institutional Ethics Committee (Approval number: 2022/133). The study protocol complied with the principles outlined in the Declaration of Helsinki. Written informed consent was obtained from all participants and their legal guardians.

Participants were children and adolescents aged 6 to 17 years who presented to the Child and Adolescent Psychiatry Clinic at Recep Tayyip Erdoğan University Rize Training and Research Hospital between April 2022 and December 2023. ADHD diagnosis was made in accordance with DSM-5 diagnostic criteria, supported by developmental and medical history, family reports, teacher evaluations of academic performance, clinical psychiatric interviews, and standardized psychometric assessments. Only patients who were medication-naive and scheduled to initiate methylphenidate therapy based on clinical evaluation were included. Inclusion criteria were as follows: (1) age between 6 and 17 years; (2) regular school attendance; (3) absence of comorbid psychiatric diagnoses (e.g., autism spectrum disorder, intellectual disability, depression); (4) no concurrent medications other than methylphenidate; (5) no chronic systemic illnesses (e.g., epilepsy, diabetes); (6) no ocular pathology other than refractive error, confirmed by a comprehensive ophthalmological examination.

Exclusion criteria included: (1) history of ocular surgery; (2) presence of chronic ocular (e.g., glaucoma, uveitis, amblyopia, retinal disorders) or systemic diseases (e.g., diabetes, hypertension); (3) use of contact lenses; (4) systemic medications known to affect corneal physiology; (5) refractive error > −2.00 diopters spherical equivalent (to minimize potential confounding effects of high myopia on corneal thickness and curvature parameters); (6) ongoing ADHD treatment; (7) coexisting psychiatric diagnoses beyond ADHD; (8) use of psychiatric medications other than methylphenidate. A visual representation of the participant selection and follow-up process is provided in Figure 1.

Demographic data, including age, gender, parental education levels, and school grade, were collected via a structured sociodemographic form developed by the researchers. ADHD symptom severity was assessed using the DSM-IV-Based Disruptive Behavior Disorders Screening and Rating Scale (Turgay), and psychiatric diagnoses were confirmed using the Schedule for Affective Disorders and Schizophrenia for School-Age Children—Present and Lifetime Version (K-SADS-PL, DSM-5, Turkish adaptation).

### 2.2. Behavioral Assessment Using the Turgay DSM-IV-Based Rating Scale

The Turgay DSM-IV-Based Disruptive Behavior Disorders Rating Scale was utilized to evaluate ADHD-related symptoms. This validated instrument consists of 41 items across four subscales: attention deficit (9 items), hyperactivity/impulsivity (9 items), oppositional defiant disorder (8 items), and conduct disorder (1 item). Each item is scored on a 4-point Likert scale ranging from 0 (Never) to 3 (Very Often). The Turkish validation of the scale was performed by Turgay et al. [14,15]. In the evaluation of the scale, scores of 0 and 1 are considered negative while scores of 2 and 3 are considered positive. In assessments conducted with this inventory, it is accepted that individuals who meet at least 6 criteria in the attention deficit section and/or the hyperactivity–impulsivity section meet the diagnostic criteria for ADHD. Higher scores indicate more severe psychopathology.

### 2.3. Ophthalmological Examination

All ADHD patients underwent a comprehensive ophthalmological evaluation prior to treatment initiation. This included best-corrected visual acuity (BCVA), intraocular pressure (IOP) measurement, anterior segment examination via slit-lamp biomicroscopy, and dilated fundus examination.

Corneal topography was conducted using the Sirius device (CSO Inc., Florence, Italy), while specular microscopy was performed with the Tomey EM-4000 (Tomey GmbH, Nishi-Ku, Nagoya, Japan). To ensure measurement precision and minimize intra-observer variability, all procedures were conducted by a single experienced technician with expertise in corneal imaging. Because specular microscopy assesses a limited area of the corneal endothelium, three consecutive technically acceptable scans were obtained and averaged for each participant. Intra-observer reliability was assessed by calculating coefficients of variation (CV) for the three consecutive measurements from each participant. Mean CV values were 2.1% for endothelial cell density and 1.4% for central corneal thickness, both within established acceptable limits for specular microscopy (ECD < 3%, CCT < 2%).

The following parameters were evaluated: Endothelial Cell Density (ECD): total number of endothelial cells in the field of view; Average Cell Area (AVE); Maximum and Minimum Cell Area (CAmax, CAmin); Standard Deviation (SD); Coefficient of Variation (CV = SD/AVE × 100): indicator of polymegethism; Hexagonality Index (6A): percentage of hexagonal cells; Central Corneal Thickness (CCT).

Measurements were repeated after six months of continuous methylphenidate use. The control group was selected from children and adolescents presenting to the Ophthalmology Outpatient Clinic with no ocular pathology other than refractive errors (myopia, hyperopia, astigmatism). Control group participants were chosen from individuals with no prior history of child psychiatry consultation. All control participants underwent systematic behavioral screening using the validated Turgay DSM-IV-Based Disruptive Behavior Disorders Screening and Rating Scale (Parent Form). The completed scales were evaluated by a child and adolescent psychiatrist, and participants scoring below clinical thresholds were designated as the control group. The same exclusion criteria established for the ADHD group were applied to control group selection. Their data were used for between-group comparisons.

### 2.4. Statistical Analysis

All statistical analyses were conducted using IBM SPSS Statistics version 26 (IBM Corp., Armonk, NY, USA). The Kolmogorov–Smirnov test was used to assess data normality. For intergroup comparisons, independent samples *t*-tests were used for normally distributed data, and Mann–Whitney U tests were used for non-normally distributed variables. Pearson’s chi-square test was used to compare categorical variables, or Fisher’s exact test when expected cell frequencies were <5.

Within-group comparisons were performed using paired *t*-tests (for normally distributed variables) or Wilcoxon signed-rank tests (for non-normal data). Effect sizes were calculated using Cohen’s d. Correlations between non-parametric variables such as Turgay scores and ophthalmologic measurements were evaluated using Spearman’s rho correlation coefficient. Quantitative data were reported as mean ± standard deviation or median (min–max), while categorical data were presented as frequencies and percentages. A *p*-value < 0.05 was considered statistically significant.

## 3. Results

A total of 100 children and adolescents diagnosed with ADHD (47 males, 53 females) and 100 healthy control participants (45 males, 55 females) were included in the final analysis. The mean age was 10.02 years in the ADHD group and 10.24 years in the control group, with no statistically significant difference observed between groups (*p* = 0.496). All participants received either intermediate-acting or extended-release methylphenidate at doses ranging from 0.3 to 1.1 mg/kg/day. Dosing was individualized based on clinical response and adjusted as needed throughout the 6-month treatment period, following standard pediatric ADHD guidelines.

Sociodemographic characteristics are summarized in Table 1. Among all variables examined, a significant difference was noted in the distribution of paternal education levels between the two groups (*p* = 0.002). Specifically, the proportion of fathers with high school education was higher in the control group (52%) compared to the ADHD group (26%).

Baseline ophthalmological parameters are presented in Table 2. The mean K average (K AVG) was significantly lower in the ADHD group (43.07) compared to the control group (43.87) (*p* < 0.001). Similarly, anterior and posterior corneal curvature asymmetry values differed significantly between groups. The median anterior curvature asymmetry (Asym F) was −1.57 in the ADHD group and 0.14 in the control group (*p* < 0.001), while the median posterior curvature asymmetry (Asym B) was −0.24 in the ADHD group and 0.04 in the control group (*p* < 0.001). 

Comparisons of ocular measurements at baseline and six months after methylphenidate treatment within the ADHD group are summarized in Table 3. Central corneal thickness (CCT) increased significantly from a median of 538 µm at baseline to 548 µm at follow-up (*p* < 0.001). Spherical equivalent (SE) also showed a statistically significant change over the same period (*p* < 0.001). Endothelial cell density (ECD) decreased significantly from a median value of 2555.5 to 2348 cells/mm^2^ (*p* < 0.001), while average cell area (AVE) increased from 361.5 to 371 µm^2^ (*p* < 0.001). Standard deviation (SD) and coefficient of variation (CV) of cell area both increased significantly (*p* < 0.001). The CV rose from a median of 39 to 42. Maximum cell area (CA-MAX) increased from a mean of 88.49 to 84.81 µm^2^ (*p* < 0.001), while minimum cell area (CA-MIN) also showed a statistically significant difference (*p* < 0.001).

The associations between ADHD symptom scores and ocular parameters are presented in Table 4. A weak but statistically significant positive correlation was identified between inattention scores and endothelial cell density (r = 0.222, *p* = 0.026). A similarly weak positive correlation was found between inattention scores and cell count (NUM) (r = 0.241, *p* = 0.016). Additionally, a weak negative correlation was observed between inattention scores and corneal volume (r = −0.248, *p* = 0.013).

## 4. Discussion

This study demonstrated that methylphenidate use in pediatric patients was associated with alterations in CCT and multiple corneal endothelial morphometric parameters. Specifically, post-treatment evaluations revealed a decrease in ECD and increases in CCT, CV, CA-MIN, CA-MAX, and 6A. The observed 207 cells/mm^2^ reduction in endothelial cell density represents an 8.1% decrease from baseline values, a change that exceeds normal measurement variability [16] and suggests measurable cellular response to chronic stimulant exposure. For context, normal pediatric values for ECD range from 3000 to 3500 cells/mm^2^ in children aged 6–17 years, while normal CCT values range from 540 to 570 μm in this age group [17]. These findings suggest that methylphenidate may be associated with alterations in corneal endothelial homeostasis at a subclinical level.

The literature on this topic is limited and includes some contradictory findings. For instance, in a prospective observational study by Koc et al., no significant changes in endothelial cell morphology were detected after six months of methylphenidate treatment in a similar pediatric cohort [6]. Discrepancies between our findings and those of prior studies may arise from methodological variations, including differences in sample selection, follow-up adherence, exclusion criteria, and sensitivity of the measurement parameters. Notably, our study utilized high-sensitivity morphometric indices such as CV, AVG, and SD, which may have enhanced our ability to detect subtle subclinical changes [16]. Such methodological differences highlight the need for standardized protocols and multicenter trials to delineate the ocular effects of psychostimulants more clearly.

Regarding intraocular pressure, no significant difference was observed between the ADHD and control groups at baseline. However, after six months of methylphenidate treatment, a statistically significant increase in IOP was noted in the ADHD group (*p* < 0.001). Although this increase remained within clinically acceptable limits, it may reflect physiological modulation of aqueous humor dynamics through enhanced sympathetic activity. Elevated dopaminergic and adrenergic stimulation is known to increase aqueous humor production and may reduce outflow via the trabecular meshwork, potentially contributing to increased IOP [11,18]. These findings suggest that IOP should be monitored periodically in pediatric patients receiving long-term stimulant therapy.

Although methylphenidate is among the first-line agents for ADHD treatment, the long-term effects of its systemic use on ocular structures, particularly the corneal endothelium, remain insufficiently characterized [12]. The pharmacological action of methylphenidate—through inhibition of dopamine and norepinephrine reuptake—is not limited to the central nervous system but may extend to peripheral tissues, including ocular structures, via the autonomic nervous system [19]. Increased sympathetic activity may modulate pupil size, IOP, and ocular perfusion, primarily through adrenergic receptor stimulation in the iris and ciliary body [20]. Given the high metabolic sensitivity of the cornea, these neuro-autonomic influences may result in structural or functional changes, even in the absence of overt symptoms [21].

In addition to these systemic neuropharmacological effects, methylphenidate-associated corneal endothelial changes may also involve direct cellular mechanisms, particularly oxidative stress-mediated damage. Corneal endothelial cells are highly susceptible to oxidative stress due to their high metabolic activity and limited antioxidant capacity, a characteristic they share with certain retinal cell types [22]. Methylphenidate has been demonstrated to induce oxidative stress in various tissues and cell types through increased reactive oxygen species production and depletion of cellular antioxidant reserves [18,23]. Given methylphenidate’s systemic delivery and pharmacokinetic properties, the drug may reach therapeutic concentrations in the aqueous humor, potentially increasing local oxidative stress levels that directly compromise corneal endothelial cellular integrity [24,25]. This oxidative mechanism likely acts synergistically with autonomic nervous system modulation, providing a more comprehensive explanation for the observed morphometric alterations.

The subclinical corneal endothelial changes observed in our study raise important questions regarding their potential evolution with prolonged methylphenidate exposure. While our 6-month observation period demonstrated measurable alterations in endothelial cell density and morphometric parameters, the long-term trajectory of these changes remains unclear. Several scenarios are theoretically possible: these changes may represent transient physiological adaptations that stabilize with continued treatment, early manifestations of progressive endothelial dysfunction, or cumulative effects that plateau at subclinical levels without further deterioration.

Drawing parallels from other pharmacological contexts, chronic exposure to medications affecting sympathetic tone has been associated with progressive tissue changes. For instance, long-term use of certain glaucoma medications has been linked to conjunctival and corneal surface alterations that worsen over time [26,27]. However, the corneal endothelium’s limited regenerative capacity in humans suggests that any initial damage may be irreversible, potentially leading to gradual accumulation of cellular dysfunction over years of treatment [28,29].

Conversely, the observed changes might represent adaptive responses to altered autonomic tone, potentially stabilizing once physiological equilibrium is achieved. The absence of clinical symptoms in our cohort suggests that current endothelial reserve remains adequate, though this may change with extended exposure or additional stressors such as contact lens wear, ocular surgery, or age-related decline [30]. Longitudinal studies with extended follow-up are essential to distinguish between these possibilities and inform evidence-based monitoring protocols.

### 4.1. Clinical Acceptability and Risk–Benefit Analysis

The clinical acceptability of our findings must be evaluated within the broader context of ADHD treatment. While we observed statistically significant corneal endothelial changes, these alterations remained subclinical and within normal pediatric ranges. Post-treatment endothelial cell density (2348 cells/mm^2^) stayed well above the critical threshold (1500–2000 cells/mm^2^) associated with corneal dysfunction [28,29], indicating adequate endothelial reserve.

The established risks of untreated ADHD are substantial and well-documented, including academic failure (80–90% of patients) [31], increased accident rates (36–58% higher) [32], and elevated psychiatric comorbidity risks [33]. Conversely, methylphenidate therapy demonstrates robust efficacy with significant improvements in core symptoms, academic performance, and safety outcomes [34].

These morphometric findings do not constitute contraindications to methylphenidate therapy. The substantial benefits of ADHD treatment significantly outweigh the theoretical long-term corneal risks identified in our study. However, these results support the implementation of enhanced ophthalmologic surveillance protocols [35].

### 4.2. Monitoring Recommendations and Clinical Implementation

Based on these findings and the principles of pediatric pharmacovigilance, we propose an evidence-informed approach to ophthalmologic surveillance. Baseline comprehensive ophthalmologic evaluation, including specular microscopy and intraocular pressure measurement, should be considered prior to methylphenidate initiation, particularly in children requiring extended treatment duration.

Follow-up intervals should be individualized based on clinical judgment and patient-specific risk stratification. For standard-risk patients with normal baseline corneal parameters, we recommend ophthalmologic evaluations at 6-month intervals during the first two years, followed by annual assessments thereafter. High-risk patients, defined as those with baseline endothelial cell density below 2500 cells/mm^2^, central corneal thickness exceeding 600 μm, or history of corneal pathology, warrant 3-month intervals initially, followed by 6-monthly evaluations.

Clinical alert thresholds requiring treatment reassessment include endothelial cell density decline exceeding 15% from baseline, development of visual symptoms, or sustained intraocular pressure elevation above 21 mmHg. Treatment decisions should maintain individualized assessment protocols while prioritizing optimal ADHD therapeutic outcomes [36].

This study has several methodological limitations that should be acknowledged. First, the observational case–control design without randomization may have introduced selection bias, as participants were not randomly assigned to treatment and control groups. Additionally, the absence of blinding procedures for examining personnel may represent a potential source of observer bias during corneal measurements and assessments. The single-center nature of the study may also limit the generalizability of our findings to broader pediatric populations.

The relatively short follow-up period of six months was selected to balance methodological feasibility with preliminary safety assessment in children and adolescents. However, this duration may not fully capture the long-term ocular effects of methylphenidate and does not allow for a clear distinction between progressive versus transient changes. Future studies should therefore extend follow-up to 12–24 months to better clarify whether observed alterations represent cumulative damage or temporary physiological adaptations.

Another limitation is the absence of a non-treated ADHD comparison group. Ethical considerations precluded maintaining ADHD-diagnosed participants without treatment for an extended period, as stimulant therapy was clinically indicated for all patients. This restricts our ability to differentiate changes associated with methylphenidate use from those potentially related to the underlying ADHD condition itself. Future research may address this issue through naturalistic designs, comparisons of different treatment modalities, or carefully supervised medication holiday protocols.

Finally, patients with refractive errors greater than −2.00 diopters spherical equivalent were excluded to minimize the potential confounding effects of high myopia on corneal parameters. While this strengthened internal validity, it may also reduce the external applicability of our findings to the broader pediatric ADHD population, where refractive errors are relatively frequent.

Taken together, these limitations underscore the need for future multicenter studies with larger samples, extended follow-up periods, and randomized controlled trial designs to definitively establish the long-term ocular safety profile of methylphenidate in pediatric populations.

The findings of this study extend beyond the scope of pediatric ophthalmology and carry significant implications for the clinical management of ADHD. The relationship between ADHD, visual function, and academic achievement represents a complex interplay that warrants careful consideration. Children with ADHD already face challenges with sustained attention and visual processing, and any additional visual strain from medication-related corneal changes could compound these difficulties [37,38].

The observed subclinical corneal endothelial changes carry potential implications that extend beyond morphometric measurements to functional visual performance in pediatric populations. While our study did not directly assess visual acuity or contrast sensitivity, the documented alterations in endothelial cell density and morphometric parameters may theoretically impact visual quality through subtle changes in corneal transparency and optical properties [39,40]. Children with compromised corneal endothelial function may experience increased susceptibility to visual symptoms during demanding near-work activities, such as reading and computer use, potentially affecting academic performance and classroom attention [41]. Furthermore, the observed increase in intraocular pressure, while remaining within normal limits, may contribute to subtle alterations in visual comfort during prolonged visual tasks, potentially manifesting as eye strain, fatigue, or difficulty with sustained near vision activities that are crucial for academic success [42].

These considerations emphasize the importance of comprehensive ophthalmologic evaluation not only for safety monitoring but also for optimizing visual function to support educational outcomes in children receiving stimulant therapy. Early detection and management of any visual complications may prevent secondary impacts on academic performance and overall quality of life.

While these findings warrant attention, it is important to contextualize the observed corneal changes within the broader therapeutic framework of ADHD management. Untreated ADHD carries significant risks, including academic failure, social difficulties, increased accident rates, and long-term psychiatric comorbidities. Therefore, these preliminary corneal findings should not be interpreted as contraindications to methylphenidate therapy, but rather as indicators for enhanced ophthalmologic surveillance. Clinical decision-making should involve individualized risk assessment, considering factors such as treatment response, alternative medication options, baseline ocular health, and family preferences, while maintaining the primary therapeutic goal of optimizing ADHD symptom management. The observation of structural changes in the corneal endothelium within six months of initiating methylphenidate treatment necessitates comprehensive ophthalmologic examination prior to treatment initiation, particularly in children requiring long-term medication use. This examination should be supported by baseline specular microscopy and IOP measurements, repeated every six months during the first two years, and annually thereafter.

It should be remembered that mild ocular discomfort or visual complaints in children using methylphenidate may represent early indicators of endothelial dysfunction. Therefore, child psychiatrists should collaborate with ophthalmologists when encountering such complaints. Additionally, consideration of alternative treatment options is warranted in children with existing corneal pathology or those at risk for endothelial damage. The integration of these findings into future pediatric ADHD treatment guidelines is of critical importance for improving patient safety.

### 4.3. Future Studies

Our findings underscore critical knowledge gaps requiring systematic investigation. Prospective, multicenter studies with extended follow-up periods (minimum 24 months) are essential to establish evidence-based monitoring intervals, validate clinically meaningful thresholds for corneal morphometric changes, and define risk stratification criteria specific to pediatric stimulant therapy. Until such validation studies provide empirical guidance, monitoring decisions should remain individualized and guided by clinical judgment rather than predetermined algorithms.

Future research should address several critical gaps in our understanding of stimulant-associated ocular effects in pediatric populations.

Study Design and Methodology: Large-scale, multicenter prospective studies with extended follow-up periods (minimum 2–3 years) are essential to determine whether the observed corneal endothelial changes represent transient adaptive responses or progressive degenerative processes. Dose–response relationship studies examining different methylphenidate formulations and dosing regimens would help establish safety thresholds and optimize treatment protocols. Additionally, comparative studies evaluating the ocular effects of alternative ADHD medications (amphetamines, atomoxetine, guanfacine) could inform evidence-based treatment selection strategies.

Advanced Imaging and Technology: Mechanistic research investigating the underlying pathophysiology of methylphenidate-associated corneal changes is critically needed. Studies incorporating advanced imaging modalities such as anterior segment optical coherence tomography, confocal microscopy, and corneal biomechanical assessment could provide deeper insights into the structural and functional consequences of chronic stimulant exposure. Longitudinal investigation of aqueous humor dynamics, including detailed analysis of production and outflow mechanisms, would enhance our understanding of IOP elevation patterns observed in our cohort.

Clinical Implementation: Furthermore, development and validation of standardized screening protocols for ophthalmologic monitoring in stimulant-treated children represents a priority research area. Such protocols should incorporate age-appropriate examination techniques, specify optimal screening intervals, and define clear criteria for treatment modification or discontinuation based on ocular findings. Health economic evaluations assessing the cost-effectiveness of routine ophthalmologic monitoring versus reactive management of complications would inform healthcare policy decisions and clinical practice guidelines for this vulnerable population.

Special Populations and Risk Factors: Age-stratified analyses are warranted to evaluate whether stimulant-associated ocular changes differ across key developmental windows (e.g., early childhood vs. adolescence), potentially guiding age-specific monitoring protocols. Exploration of genetic or pharmacogenomic factors that predispose certain children to ocular side effects may help identify high-risk individuals and personalize ADHD treatment plans.

Future Technological Applications: Incorporating functional assessments such as contrast sensitivity testing, accommodative response, and visual attention paradigms could bridge structural findings with neuro-visual performance in ADHD. Real-world data analyses using electronic health records or school-based screening databases could complement prospective studies and support population-level surveillance. Artificial intelligence-assisted image analysis tools may enhance detection of subtle corneal changes and enable scalable screening strategies in clinical settings.

These comprehensive research initiatives will ultimately contribute to the development of evidence-based guidelines for safe and effective ADHD management in pediatric populations.

## 5. Conclusions

This study demonstrates that long-term methylphenidate treatment in pediatric patients with ADHD is associated with measurable alterations in corneal endothelial morphology and intraocular pressure. Notably, reductions in endothelial cell density and increases in morphometric variability suggest potential subclinical stress on corneal tissue integrity. Although these changes did not reach overt clinical thresholds, their cumulative nature underscores the importance of routine ophthalmologic monitoring in children receiving chronic stimulant therapy. Our findings contribute to the growing body of evidence on the peripheral ocular effects of psychostimulants and highlight the need for larger, multicenter studies with extended follow-up to better understand the neuro-ophthalmologic interface in pediatric psychopharmacology.

## Figures and Tables

**Figure 1 diagnostics-15-02368-f001:**
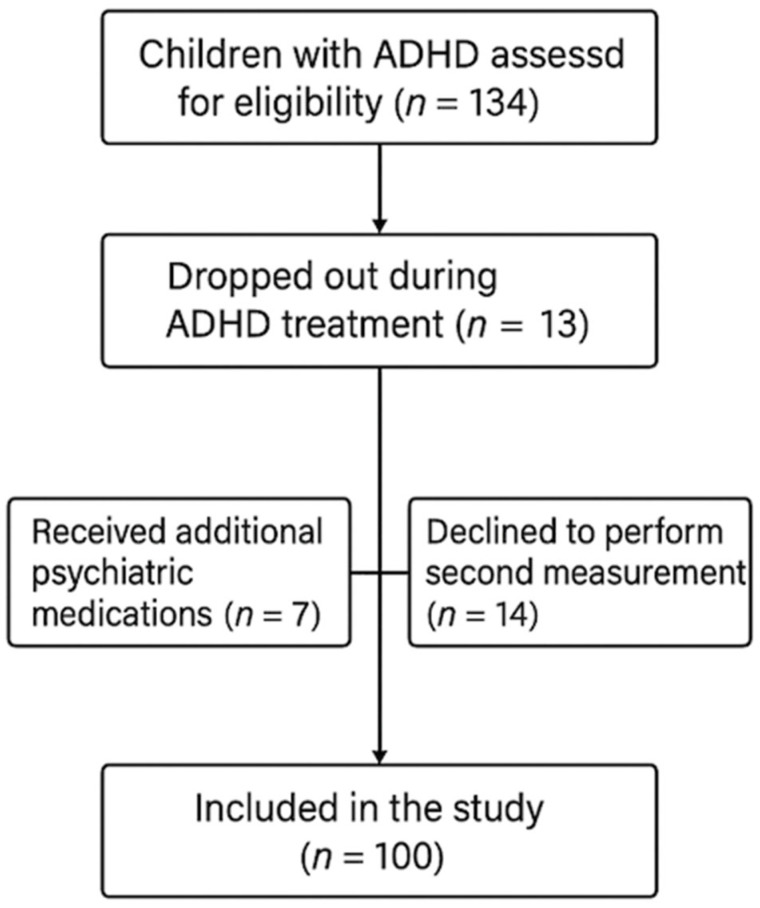
Flowchart showing inclusion, exclusion, and follow-up of ADHD and control participants over the study period.

**Table 1 diagnostics-15-02368-t001:** Descriptive statistics of demographic characteristics.

	Group		Test Statistics	*p*
	Control (*n* = 100)	Patient (*n* = 100)		
Age	10.24 ± 2.12	10.02 ± 2.4	0.681	0.496 ^t^
Weight-Kg	43 (21–58)	39 (21–58)	−1.412	0.158 ^m^
Number of Siblings				
1	12 (12)	7 (7)	1.779	0.613 ^f^
2	54 (54)	54 (54)		
3	32 (32)	37 (37)		
4	2 (2)	2 (2)		
Siblings Order				
1	63 (63)	67 (67)	2.510	0.311 ^f^
2	33 (33)	25 (25)		
3	4 (4)	8 (8)		
Mother’s Education Status				
Primary Education	23 (23)	28 (28)	1.465	0.702 ^f^
High School	36 (36)	29 (29)		
College/University	32 (32)	35 (35)		
Graduate/Postgraduate	9 (9)	8 (8)		
Father’s Education Status				
Primary Education	4 (4) ^a^	9 (9) ^a^	15.487	0.002 ^f^
High School	52 (52) ^a^	26 (26) ^b^		
College/University	36 (36) ^a^	48 (48) ^a^		
Graduate/Postgraduate	8 (8) ^a^	17 (17) ^a^		
Turgay				
Inattention	7 (5–9)	23 (12–32)	−12.322	<0.001
Hyperactivity/Impulsivity	6 (0–9)	12 (0–26)	−8.164	<0.001
ODD	5 (0–9)	8 (0–23)	−5.114	<0.001
CD	0 (0–16)	0 (0–10)	−0.958	0.338

^t^: Independent two sample *t* test, ^m^: Mann–Whitney U test, ^f^: Fisher’s exact test, ^a,b^: No difference between groups with the same letter (Bonferroni corrected Z test), ODD: Oppositional Defiant Disorder; CD: Conduct Disorder.

**Table 2 diagnostics-15-02368-t002:** Descriptive statistics of clinical characteristics.

	Group		Test Statistics	*p*	Effect Size
	Control (*n* = 100)	Patient (*n* = 100)			
IOP	15 (13–18)	16 (15–19)	−4.292	0.345 ^m^	d_c_ = 2.653
CCT	537 (507–566)	538 (523–576)	−5.973	0.214 ^m^	d_c_ = 0.932
SE	−0.25 (−13–1.5)	−0.75 (−1.25–0.75)	−2.144	0.132 ^m^	d_c_ = 0.307
AL	23.5 (22.96–24.24)	23.69 (22.96–24.24)	−1.012	0.311 ^m^	d_c_ = 0.143
AD	3.1 (2.88–3.41)	3.12 (2.88–3.41)	−0.804	0.422 ^m^	d_c_ = 0.114
ACD	3.43 (2.96–3.89)	3.25 (2.96–3.89)	−1.812	0.070 ^m^	d_c_ = 0.258
LT	3.56 (3.26–3.94)	3.56 (3.26–3.94)	−0.042	0.967 ^m^	d_c_ = 0.006
WTW	12.15 (11.52–12.88)	12.13 (11.52–12.88)	−0.159	0.874 ^m^	d_c_ = 0.022
ECD	2574.5 (1243–2941)	2595.5 (2461–2760)	−2.304	0.061 ^m^	d_c_ = 0.330
NUM	265.5 (67–302)	278 (247–313)	−7.973	0.255 ^m^	d_c_ = 1.365
AVG	362 (340–468)	361.5 (354–368)	−0.594	0.555 ^m^	d_c_ = 0.084
SD	116 (99–196)	116 (99–136)	−0.108	0.914 ^m^	d_c_ = 0.015
CV	39 (32–47)	39 (35–47)	−0.009	0.993 ^m^	d_c_ = 0.001
CA-MAX	811.19 ± 60.45	808.49 ± 58.42	0.321	0.748 ^t^	d_c_ = 0.045
CA-MIN	99 (4–213)	99.5 (82–128)	−0.377	0.707 ^m^	d_c_ = 0.053
6A	49 (26–62)	50 (45–54)	−6.019	0.894 ^m^	d_c_ = 1.143
Corneal Volume	56.71 (51.9–66.3)	56.57 (53.67–58.94)	−0.623	0.533 ^m^	d_c_ = 0.088
K1	43.44 (40.36–47.18)	43.28 (41.99–44.12)	−0.925	0.355 ^m^	d_c_ = 0.131
K2	42.76 (40.67–46.37)	42.76 (41.76–44.68)	−0.078	0.938 ^m^	d_c_ = 0.011
K AVG	43.87 ± 1.59	43.07 ± 0.46	4.773	<0.001 ^t^	d_c_ = 0.675
K CYL	−0.71 (−3.09–−0.1)	−0.7 (−1.2–−0.06)	−1.548	0.122 ^m^	d_c_ = 0.220
HVID	12.05 (11.1–12.88)	11.94 (11.37–12.79)	−0.683	0.494 ^m^	d_c_ = 0.097
Apex Curvature	44.69 (43.87–54.23)	44.68 (43.87–54.23)	−0.363	0.717 ^m^	d_c_ = 0.051
Curvature ASYM F	0.14 (0–0.75)	−1.57 (−1.63–1.61)	−12.009	<0.001 ^m^	d_c_ = 3.216
Curvature ASYM B	0.04 (0–1.06)	−0.24 (−0.28–−0.21)	−12.305	<0.001 ^m^	d_c_ = 3.531
Iridocorneal Angle	47 (35–55)	49 (42–57)	−2.859	0.213 ^m^	d_c_ = 1.109

^t^: Independent two sample *t* test, ^m^: Mann–Whitney U test; d_c_: Cohen’s d. SE: spherical equivalent; CCT: central corneal thickness; WTW: White to White; ECD: endothelial cell density; NUM: number of cells; AVG: average cell area; CA-MAX: maximum cell area; CA-MIN: minimum cell area; SD: standard deviation; CV: coefficient of variation in cell area. 6A: the percentage of hexagonal endothelial cells; K1: Keratometry flat K2: Keratometry steep K Avg: Keratometry average K CYL: Keratometry cylindric; HVID: horizontal visible iris diameter.

**Table 3 diagnostics-15-02368-t003:** Examination of the difference between initial and final measurements in the patient group.

	First Measurement (*n* = 100)	Measurement After 6 Months (*n* = 100)	Test ist.	*p*	R^2^	Cohen’s d
CCT	538 (523–576)	548 (523–586)	−8.678	<0.001 ^w^	0.825	3.519
IOP	15 (13–18)	18 (15–21)	−11.292	<0.001 ^w^	0.786	3.127
SE	−0.75 (−1.25–0.75)	−1.0 (−1.5–−0.75)	−8.619	<0.001 ^w^	0.575	3.340
AL	23.69 (22.96–24.24)	23.8 (22.91–24.44)	−2.007	0.065 ^w^	0.969	0.410
AD	3.12 (2.88–3.41)	3.11 (2.87–3.28)	−2.601	0.017 ^w^	0.892	0.539
ACD	3.25 (2.96–3.89)	3.26 (3.01–3.89)	−1.257	0.209 ^w^	0.995	0.253
LT	3.56 (3.26–3.94)	3.57 (3.25–3.96)	−2.991	0.023 ^w^	0.986	0.627
WTW	12.13 (11.52–12.88)	12.12 (11.55–12.86)	−0.52	0.603 ^w^	0.997	0.124
ECD	2595.5 (2461–2760)	2348 (2199–2590)	−8.682	<0.001 ^w^	0.779	3.499
NUM	278 (247–313)	278 (247–313)	−0.528	0.598 ^w^	−0.058	0.106
AVG	361.5 (354–368)	371 (359–378)	−8.704	<0.001 ^w^	0.844	3.536
SD	116 (99–136)	120.96 ± 9.39	−12.065	<0.001 ^e^	0.920	1.229
CV	39 (35–47)	42 (37–49)	−9.069	<0.001 ^w^	0.979	4.305
CA-MAX	808.49 ± 58.42	834.81 ± 60.61	−20.838	<0.001 ^e^	0.978	2.148
CA-MIN	99.5 (82–128)	114 (94–136)	−8.688	<0.001 ^w^	0.880	3.509
6A	50 (45–54)	48 (43–52)	−8.981	<0.001 ^w^	0.941	4.084
Corneal Volume	56.57 (53.67–58.94)	56.52 (53.72–58.99)	−2.168	0.130 ^w^	0.997	0.444
K1	43.28 (41.99–44.12)	43.32 (41.87–44.16)	−1.256	0.209 ^w^	0.937	0.353
K2	42.26 (41.66–45.34)	42.76 (41.78–44.64)	−2.067	0.139 ^w^	0.982	0.423
K AVG	43.07 (42.23–45.12)	43.05 (42.07–44.41)	−2.363	0.018 ^w^	0.963	0.486
K CYL	−0.7 (−1.2–−0.06)	−0.65 ± 0.33	−1.174	0.243 ^e^	0.579	0.117
HVID	11.94 (11.37–12.79)	11.96 (11.34–12.77)	−1.061	0.289 ^w^	0.921	0.213
Apex Curvature	44.68 (43.87–54.23)	44.67 (43.71–54.34)	−0.891	0.373 ^w^	0.995	0.179
Curvature Asym-F	−1.57 (−1.63–1.61)	−1.56 (−1.65–−1.46)	−0.202	0.840 ^w^	0.059	0.040
Curvature Asym-B	−0.24 (−0.28–−0.21)	−0.24 (−0.29–−0.19)	−3.212	0.021 ^w^	0.549	0.678
Irıdocorneal Angle	49 (42–57)	48.5 (42–58)	−0.943	0.346 ^w^	0.855	0.189

^w^: Wilcoxon test, ^e^: Paired two-sample *t* test; SE: spherical equivalent; IOP: intraocular pressure; CCT: central corneal thickness; WTW: White to White; ECD: endothelial cell density; NUM: number of cells; AVG: average cell area; CA-MAX: maximum cell area; CA-MIN: minimum cell area; SD: standard deviation; CV: coefficient of variation in cell area. 6A: the percentage of hexagonal endothelial cells; K1: Keratometry flat K2: Keratometry steep K Avg: Keratometry average K CYL: Keratometry cylindric; HVID: horizontal visible iris diameter.

**Table 4 diagnostics-15-02368-t004:** Examination of the relationship between Turgay scale scores and changes in eye measurements in the patient group.

Difference *		Weight-Kg	A-Turgay–Carelessness	Turgay-Hyperactivity	Turgay-Kokgb	Turgay-Db
IOP	r	0.097	0.049	−0.057	0.143	0.122
	*p*	0.335	0.626	0.576	0.156	0.228
SE	r	0.025	0.213	0.073	0.095	0.022
	*p*	0.804	0.033	0.472	0.345	0.828
AL	r	0.002	0.062	0.018	−0.241	−0.073
	*p*	0.984	0.541	0.860	0.016	0.472
CCT	r	0.101	−0.156	−0.123	−0.131	−0.241
	*p*	0.318	0.121	0.224	0.194	0.016
AD	r	−0.018	0.092	0.112	−0.053	0.054
	*p*	0.856	0.360	0.269	0.600	0.593
ACD	r	0.069	−0.108	−0.181	−0.086	−0.078
	*p*	0.495	0.287	0.072	0.396	0.441
LT	r	−0.068	0.107	0.127	−0.027	−0.173
	*p*	0.499	0.290	0.209	0.788	0.084
WTW	r	0.010	−0.167	−0.046	0.023	0.025
	*p*	0.923	0.096	0.652	0.817	0.806
ECD	r	−0.040	0.222	−0.034	−0.036	−0.086
	*p*	0.695	0.026	0.741	0.723	0.398
NUM	r	0.005	0.241	0.012	0.059	0.111
	*p*	0.958	0.016	0.908	0.559	0.271
AVG	r	−0.146	−0.005	−0.091	−0.037	0.039
	*p*	0.147	0.957	0.366	0.715	0.701
SD	r	0.092	0.079	−0.063	0.059	−0.031
	*p*	0.365	0.433	0.533	0.561	0.757
CV	r	0.021	0.111	−0.043	0.061	0.136
	*p*	0.839	0.274	0.671	0.546	0.178
CA-MAX	r	0.099	0.050	0.056	0.074	−0.020
	*p*	0.325	0.623	0.578	0.464	0.841
CA-MIN	r	−0.013	−0.124	−0.123	−0.079	−0.065
	*p*	0.895	0.221	0.223	0.437	0.521
6A	r	0.113	−0.109	0.191	−0.030	0.045
	*p*	0.264	0.280	0.057	0.768	0.660
Corneal Volume	r	−0.083	−0.248	0.061	−0.062	0.072
	*p*	0.413	0.013	0.546	0.540	0.474
K1	r	0.100	0.146	0.096	−0.041	0.117
	*p*	0.320	0.149	0.340	0.688	0.245
K2	r	0.073	0.025	0.041	−0.199	−0.033
	*p*	0.470	0.802	0.685	0.048	0.747
K AVG	r	0.140	0.205	0.174	−0.068	0.160
	*p*	0.164	0.041	0.083	0.500	0.112
K CYL	r	−0.011	−0.296	−0.146	0.061	−0.189
	*p*	0.915	0.003	0.148	0.549	0.059
HVID	r	0.075	0.085	0.096	−0.040	0.041
	*p*	0.460	0.402	0.344	0.691	0.689
Apex Curvature	r	−0.084	0.002	0.019	−0.044	−0.049
	*p*	0.408	0.981	0.852	0.661	0.630
Curvature Asym-F	r	0.055	−0.016	0.028	0.221	0.021
	*p*	0.586	0.873	0.781	0.027	0.835
Curvature Asym-B	r	−0.025	−0.001	−0.033	−0.005	−0.072
	*p*	0.808	0.995	0.747	0.958	0.474
Irıdocorneal Angle	r	−0.088	0.035	0.038	−0.079	0.082
	*p*	0.386	0.727	0.710	0.435	0.420

r: Spearman’s rho coefficient of correlation, * difference between first and last measurement. IOP: intraocular pressure; SE: spherical equivalent; CCT: central corneal thickness; WTW: White to White; ECD: endothelial cell density; NUM: number of cells; AVG: average cell area; CA-MAX: maximum cell area; CA-MIN: minimum cell area; SD: standard deviation; CV: coefficient of variation in cell area. 6A: the percentage of hexagonal endothelial cells; K1: Keratometry flat K2: Keratometry steep K Avg: Keratometry average K CYL: Keratometry cylindric; HVID: horizontal visible iris diameter.

## Data Availability

The original contributions presented in this study are included in the article. Further inquiries can be directed to the corresponding author.

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
