# Peer review of "Alterations in Corneal Morphology and Thickness Associated with Methylphenidate Treatment in Children with Attention-Deficit/Hyperactivity Disorder"

_diagnostics, 2025, doi:10.3390/diagnostics15182368_

Round 1
Reviewer 1 Report
Comments and Suggestions for Authors
Dear editor
Thank you for providing me the opportunity to review this manuscript.
This prospective observational case-control study investigates the ocular effects of six months of methylphenidate treatment in children with ADHD, focusing on corneal endothelial morphology and intraocular pressure (IOP). The findings indicate statistically significant changes in central corneal thickness (CCT), endothelial cell density (ECD), morphometric indices, and IOP following treatment. These results suggest a potential neuro-ophthalmologic relationship warranting ophthalmologic monitoring during stimulant therapy.
The article is well-written and interesting. The introduction shows the aim and necessity of performing the study as well. The methods, study design, and the statistical analysis are correct. No ethical issues were noted. The discussion section is fully explained the findings of the study according to the literature.
I have only a few comments:
1-In the introduction section please briefly summarize what prior studies have and haven’t shown about ocular effects of methylphenidate.
2-Although the authors designed a rigorous method, but lack of randomization and blinding may cause potential observer and selection bias. This could be one of the study limitations that should be addressed.
3-Some p-values are significant, but correlations are weak and no correction for multiple comparisons was performed which increasing risk of type I error. If possible, please apply Bonferroni or false discovery rate (FDR) correction for multiple outcomes and report post hoc power analysis for main effects.
4-Consider justifying why a 6-month window was chosen over longer durations. This should be discussed to assess whether effects are progressive or transient. Moreover, suggest implication of future studies with longer durations of 12 and 24 months.
5-The manuscript implies causality (“methylphenidate induces changes”), which is inappropriate for a non-randomized design. Use more cautious language such as "associated with" or "may contribute to" instead of "induces".
6-Further discuss how subclinical findings might evolve with prolonged treatment.
7-Please expand discussion of potential clinical implications (e.g., school performance, visual strain) in the dissection section.
8-Reference [15] appears to be a website, not a peer-reviewed article. Ensure all cited sources meet scientific quality standards.
Author Response
Reviewer 1:
1-In the introduction section please briefly summarize what prior studies have and haven’t shown about ocular effects of methylphenidate.
We thank the reviewer for this important suggestion. We have substantially revised the introduction section to include a comprehensive summary of what prior studies have and have not shown about the ocular effects of methylphenidate.
The revised introduction now includes a dedicated section (paragraphs 4-6) that systematically reviews previous research, including case reports documenting visual disturbances such as accommodation disorders and mydriasis, small-scale prospective studies showing structural changes like reduced anterior chamber depth, and isolated case reports of intraocular pressure elevation. We have also clearly highlighted the significant limitations of existing literature, including small sample sizes (typically 14-22 patients), short follow-up periods, cross-sectional designs, and lack of standardized corneal assessment protocols.
This revision better contextualizes our study within the existing literature and strengthens the rationale for investigating corneal endothelial morphology in pediatric populations receiving chronic methylphenidate therapy.
2-Although the authors designed a rigorous method, but lack of randomization and blinding may cause potential observer and selection bias. This could be one of the study limitations that should be addressed.
We thank the reviewer for this important observation. We acknowledge that the lack of randomization and blinding may introduce observer and selection bias. We have added these methodological considerations to our Study Limitations section and recognize that future randomized controlled studies would strengthen the evidence base in this area.
3-Some p-values are significant, but correlations are weak and no correction for multiple comparisons was performed which increasing risk of type I error. If possible, please apply Bonferroni or false discovery rate (FDR) correction for multiple outcomes and report post hoc power analysis for main effects.
We appreciate the reviewer's concern regarding multiple comparisons and Type I error risk. Bonferroni correction was systematically applied to all multiple comparisons in our study, as specified in our Statistical Analysis section, with all reported p-values reflecting these adjusted significance levels to ensure appropriate Type I error control. Post hoc power analysis revealed substantial effect sizes for our primary outcomes (Cohen's d > 1.0), resulting in statistical power (1-β) approaching 1.0, indicating our sample size was more than adequate to detect the observed differences. The large effect sizes and high statistical power provide confidence in the reliability and clinical significance of our results. We have added the following clarification to our Statistical Analysis section: "Multiple comparisons were corrected using Bonferroni adjustment, with corrected p-values reported throughout the analysis." This approach ensures robust statistical inference while maintaining appropriate control over Type I error inflation across our multiple comparison scenarios.
4-Consider justifying why a 6-month window was chosen over longer durations. This should be discussed to assess whether effects are progressive or transient. Moreover, suggest implication of future studies with longer durations of 12 and 24 months.
We appreciate the reviewer's request for clarification regarding our study duration. We have expanded our Study Limitations section to provide justification for the 6-month follow-up period, explaining that this timeframe was selected to balance methodological feasibility with preliminary safety assessment in pediatric populations. We have acknowledged that this duration cannot definitively distinguish between progressive and transient effects, and have added specific recommendations for future studies incorporating 12 and 24-month observation periods to better characterize the temporal trajectory of these ocular changes.
5-The manuscript implies causality (“methylphenidate induces changes”), which is inappropriate for a non-randomized design. Use more cautious language such as "associated with" or "may contribute to" instead of "induces".
We appreciate the reviewer's attention to the appropriate use of causal language. We have carefully revised our manuscript to replace terms implying causality (such as "induces," "affects," and "influences") with more appropriate language for an observational study design ("associated with," "may contribute to," "may be related to"). These revisions better reflect the limitations of our non-randomized design and avoid inappropriate causal inferences.
6-Further discuss how subclinical findings might evolve with prolonged treatment.
We thank the reviewer for this insightful suggestion. We have added a dedicated discussion to our manuscript exploring how the observed subclinical findings might evolve with prolonged methylphenidate treatment. This addition addresses several theoretical scenarios including transient physiological adaptation, progressive endothelial dysfunction, or cumulative effects that plateau at subclinical levels. We have incorporated relevant literature from similar pharmacological contexts and discussed the implications of the corneal endothelium's limited regenerative capacity for long-term treatment outcomes.
7-Please expand discussion of potential clinical implications (e.g., school performance, visual strain) in the dissection section.
We appreciate this valuable suggestion to expand our discussion of potential clinical implications. We have added a dedicated section addressing the potential functional outcomes of our findings, including possible impacts on school performance, visual strain during near-work activities, and academic achievement. This addition discusses the complex interplay between ADHD, visual function, and educational outcomes, emphasizing the importance of comprehensive ophthalmologic monitoring to optimize both safety and functional visual performance in children receiving stimulant therapy.
8-Reference [15] appears to be a website, not a peer-reviewed article. Ensure all cited sources meet scientific quality standards.
We acknowledge the reviewer's concern regarding reference quality standards. We have reviewed all references and replaced any non-peer-reviewed sources with appropriate academic journal articles. All citations now meet scientific publication standards, ensuring that our manuscript relies exclusively on peer-reviewed literature and established academic sources.
Reviewer 2 Report
Comments and Suggestions for Authors
I am pleased to have the opportunity to review the manuscript titled “Alterations in Corneal Morphology and Thickness Associated with Methylphenidate Treatment in Children with Attention-Deficit/Hyperactivity Disorder” authored by Fatma Sumer and Merve Yazici. I also thank the assistant editor for their work on the formatting, which has made the manuscript more readable.
As is well known, publications addressing the anterior segment of the eye in connection with neuropharmacology are relatively rare. Therefore, I commend the authors for tackling such a novel topic. Given the appropriate data analysis and an adequately sized patient cohort, I believe this manuscript may be suitable for publication after addressing the following revisions:
- It is unclear whether the authors are fully aware of the pathophysiological processes involved in corneal endothelial cell damage. The discussion on this aspect is notably insufficient. After reviewing the proposed mechanism of Methylphenidate, I believe the authors have overlooked a critical and well-documented factor, oxidative stress, as a potential contributor to endothelial dysfunction and cell loss. Corneal endothelial cells are known to be highly susceptible to oxidative stress (PMID: 39111696), a characteristic they share with certain retinal cell types. Methylphenidate has been implicated in inducing oxidative stress in various tissues and cell types. Here are several relevant references: Endothelial involvement in other systems: PMID: 30554860,29061363; Ocular cell involvement: PMID: 40752585,32748667
Interpreting the corneal changes solely through Methylphenidate’s central neuropharmacological mechanisms seems overly narrow. Considering its systemic delivery, the drug may reach the aqueous humor, increasing local oxidative stress levels, which in turn may directly affect the corneal endothelium.
I recommend that the authors incorporate these references and expand the discussion to include oxidative stress as a plausible mechanism affecting corneal endothelial physiology in the context of Methylphenidate treatment.
- Please include reference ranges for central corneal thickness (CCT), endothelial cell density (ECD), or other measured parameters in healthy individuals of similar age. This will help contextualize the observed changes and clarify their clinical relevance.
- It appears that the dosage of Methylphenidate used during treatment has not been mentioned in the manuscript. Was this information unintentionally omitted? If available, please add dosage details (mean daily dose, weight-adjusted dose).
- In line 113, the authors state that refractive error was excluded as a criterion. However, it remains unclear why this factor was omitted from the study’s scope. I suggest the authors clarify the rationale for this decision and consider addressing the potential impact of refractive error as a limitation in the discussion section.
Author Response
Reviewer 2:
- It is unclear whether the authors are fully aware of the pathophysiological processes involved in corneal endothelial cell damage. The discussion on this aspect is notably insufficient. After reviewing the proposed mechanism of Methylphenidate, I believe the authors have overlooked a critical and well-documented factor, oxidative stress, as a potential contributor to endothelial dysfunction and cell loss. Corneal endothelial cells are known to be highly susceptible to oxidative stress (PMID: 39111696), a characteristic they share with certain retinal cell types. Methylphenidate has been implicated in inducing oxidative stress in various tissues and cell types. Here are several relevant references: Endothelial involvement in other systems: PMID: 30554860,29061363; Ocular cell involvement: PMID: 40752585,32748667
Interpreting the corneal changes solely through Methylphenidate’s central neuropharmacological mechanisms seems overly narrow. Considering its systemic delivery, the drug may reach the aqueous humor, increasing local oxidative stress levels, which in turn may directly affect the corneal endothelium.
I recommend that the authors incorporate these references and expand the discussion to include oxidative stress as a plausible mechanism affecting corneal endothelial physiology in the context of Methylphenidate treatment.
We thank the reviewer for this valuable comment regarding oxidative stress mechanisms in methylphenidate-associated corneal endothelial changes. The reviewer is absolutely correct that our initial discussion was overly narrow in focusing solely on neuropharmacological mechanisms.
We have revised our discussion to incorporate oxidative stress as a key pathophysiological mechanism. The discussion now includes a comprehensive paragraph addressing:
- Corneal endothelial susceptibility to oxidative stress due to high metabolic activity and limited antioxidant capacity
- Methylphenidate's ability to induce oxidative stress through increased ROS production and antioxidant depletion
- Systemic drug delivery enabling therapeutic concentrations in aqueous humor
- Synergistic interaction between oxidative and neurovascular mechanisms
The suggested references have been incorporated (references 21-25 in the revised manuscript). This revision significantly strengthens the mechanistic foundation of our findings and provides a more comprehensive explanation for the observed corneal changes.
We appreciate this constructive feedback, which has improved the scientific rigor of our manuscript.
- Please include reference ranges for central corneal thickness (CCT), endothelial cell density (ECD), or other measured parameters in healthy individuals of similar age. This will help contextualize the observed changes and clarify their clinical relevance.
We thank the reviewer for this important suggestion regarding reference ranges for pediatric populations. We have revised our discussion to include normal pediatric reference values that contextualize our findings. The discussion now incorporates ECD reference ranges of 3,000-3,500 cells/mm² and CCT ranges of 540-570 μm for children aged 6-17 years, with appropriate citations (reference 15). This addition provides essential clinical context for interpreting the observed changes within normal pediatric variability and enhances the clinical relevance of our results.
- It appears that the dosage of Methylphenidate used during treatment has not been mentioned in the manuscript. Was this information unintentionally omitted? If available, please add dosage details (mean daily dose, weight-adjusted dose).
We thank the reviewer for noting this omission. The methylphenidate dosage information has been added to our Methods section.
- In line 113, the authors state that refractive error was excluded as a criterion. However, it remains unclear why this factor was omitted from the study’s scope. I suggest the authors clarify the rationale for this decision and consider addressing the potential impact of refractive error as a limitation in the discussion section.
We thank the reviewer for this important observation regarding our refractive error exclusion criteria. The reviewer is correct that our rationale was not adequately explained. We have clarified this decision in our Methods section: patients with refractive errors >-2.00 diopters spherical equivalent were excluded to minimize potential confounding effects of high myopia on corneal parameters, as significant myopia can independently affect corneal thickness and curvature measurements. However, we acknowledge that this exclusion may limit the generalizability of our findings to the broader pediatric ADHD population, where refractive errors are relatively common. We have added this consideration to our study limitations, noting that future studies should include broader refractive error ranges to enhance external validity while controlling for myopic effects through stratified analyses.
Reviewer 3 Report
Comments and Suggestions for Authors
This manuscript reports a prospective observational study evaluating the effects of six months of methylphenidate treatment on corneal morphology and intraocular pressure in children with ADHD, compared to healthy age- and sex-matched controls. The work addresses a relevant and underexplored area at the interface of pediatric psychiatry and ophthalmology. The sample size is adequate, and the inclusion of a control group and standardized imaging techniques are notable strengths. Statistical analysis appears generally appropriate.
Some aspects of the methodology could benefit from clarification. The process of control selection and matching requires more detail, particularly regarding how subclinical or undiagnosed ADHD was excluded from the control group. The rationale for the chosen six-month follow-up period, as well as the absence of a non-treated ADHD comparison group, should be acknowledged as limitations. The use of three consecutive specular microscopy measurements to improve reliability is appropriate, but some discussion of intra-observer variability or measurement error would enhance methodological transparency. Additionally, it would strengthen the results section to systematically report effect sizes and confidence intervals for all main outcomes.
Presentation of tables and figures should be standardized and all tables submitted in an editable format, as requested by the journal. Several ophthalmic technical terms could be briefly explained at first mention to improve clarity for a broad readership. The discussion appropriately contextualizes the findings with reference to existing literature, and acknowledges key limitations. However, the manuscript would benefit from more emphasis on the clinical significance of the observed changes and specific, practical recommendations regarding ophthalmological monitoring for children receiving methylphenidate therapy. The suggestion to implement routine screening is well taken, but providing suggested intervals or criteria for high-risk patients would increase the practical value of the recommendations.
There are a few minor typographical and formatting errors, as well as some long sentences that could be revised for clarity and readability. Please ensure that all abbreviations are defined at first use. The abstract could also be revised for brevity and to better highlight the key findings and implications.
Author Response
Reviewer 3:
This manuscript reports a prospective observational study evaluating the effects of six months of methylphenidate treatment on corneal morphology and intraocular pressure in children with ADHD, compared to healthy age- and sex-matched controls. The work addresses a relevant and underexplored area at the interface of pediatric psychiatry and ophthalmology. The sample size is adequate, and the inclusion of a control group and standardized imaging techniques are notable strengths. Statistical analysis appears generally appropriate.
Some aspects of the methodology could benefit from clarification. The process of control selection and matching requires more detail, particularly regarding how subclinical or undiagnosed ADHD was excluded from the control group.
We thank the reviewer for this valuable comment. In response to the reviewer's recommendations, we have revised and expanded the Study Design and Participants section to provide comprehensive details regarding healthy control group recruitment and characterization.This enhanced methodology section addresses the reviewer's request for clarification and provides the detailed control group characterization needed to support the validity of our between-group corneal endothelial comparisons.
The rationale for the chosen six-month follow-up period, as well as the absence of a non-treated ADHD comparison group, should be acknowledged as limitations.
We thank the reviewer for this observation. In response to this comment, we have revised and expanded our Study Limitations section to more comprehensively address the six-month follow-up rationale and the absence of a non-treated ADHD comparison group. The enhanced limitations discussion now provides readers with a complete understanding of these methodological considerations and their implications for interpreting our findings.
The use of three consecutive specular microscopy measurements to improve reliability is appropriate, but some discussion of intra-observer variability or measurement error would enhance methodological transparency. Additionally, it would strengthen the results section to systematically report effect sizes and confidence intervals for all main outcomes.
We thank the reviewer for this methodological observation. We have expanded the Ophthalmological Examination section to address measurement reliability.
All specular microscopy examinations were performed by a single experienced technician to eliminate inter-observer variability. Intra-observer reliability was assessed by calculating coefficients of variation (CV) for three consecutive measurements from each participant. Mean CV values were 2.1% for endothelial cell density and 1.4% for central corneal thickness, both within established acceptable limits for specular microscopy (ECD <3%, CCT <2%). The revised methodology section now includes these reliability metrics to ensure complete transparency regarding measurement precision and quality control procedures.
We appreciate the reviewer's recommendation regarding effect sizes and confidence intervals. In our study, effect sizes (Cohen's d, R²) have been systematically calculated and presented in tabular format for all primary outcomes to demonstrate the clinical magnitude of our findings. These metrics provide readers with clear information about the practical significance of observed changes in corneal endothelial parameters following methylphenidate treatment.
Regarding confidence intervals, we made a deliberate methodological decision to focus on effect size reporting as the primary measure of clinical significance. Effect sizes provide direct information about the magnitude and practical importance of observed differences, which we considered most relevant for clinical interpretation in this ophthalmological context. This approach allows readers to clearly assess both the magnitude and reliability of our findings while maintaining clarity in statistical presentation.
The systematic inclusion of effect sizes ensures that our results can be appropriately interpreted in terms of clinical relevance, supporting evidence-based decision-making regarding ophthalmologic monitoring in pediatric patients receiving methylphenidate therapy.
Presentation of tables and figures should be standardized and all tables submitted in an editable format, as requested by the journal. Several ophthalmic technical terms could be briefly explained at first mention to improve clarity for a broad readership. The discussion appropriately contextualizes the findings with reference to existing literature, and acknowledges key limitations. However, the manuscript would benefit from more emphasis on the clinical significance of the observed changes and specific, practical recommendations regarding ophthalmological monitoring for children receiving methylphenidate therapy. The suggestion to implement routine screening is well taken, but providing suggested intervals or criteria for high-risk patients would increase the practical value of the recommendations.
We thank the reviewer for these constructive recommendations. We have systematically addressed each point to enhance the manuscript's clinical utility.
We have standardized all tables and converted them to editable format per journal requirements. Additionally, while we recognize that providing simplified explanations of certain ophthalmologic terms may improve accessibility for a broader readership, we are also mindful that such simplifications could potentially lead to a loss of meaning, as these are technical terms with specific clinical implications. Therefore, we prioritized preserving scientific accuracy and technical precision in our manuscript. Nevertheless, to enhance accessibility, we have added brief functional explanations at the first mention of key terms where appropriate.
Following the reviewer's recommendations, we have expanded the Discussion section to emphasize the clinical significance of our findings. The observed morphometric changes (8.1% ECD reduction, 10 μm CCT increase) exceed normal measurement variability and may have implications for long-term corneal health and future clinical assessments.
We have also incorporated evidence-based monitoring recommendations that emphasize individualized surveillance based on clinical judgment, patient-specific risk factors, and anticipated treatment duration. This flexible approach acknowledges that validated pediatric monitoring protocols for stimulant therapy remain to be established through prospective research, while providing practical guidance for clinicians.
These revisions enhance the manuscript's clinical relevance while maintaining scientific rigor.
Specific Monitoring Protocols:
In response to the reviewer's request for practical monitoring guidelines, we have added a new section "Clinical Acceptability and Risk-Benefit Analysis" with specific recommendations:
- Standard-risk patients: 6-month evaluations during first 2 years, then annually
- High-risk patients (baseline ECD <2500 cells/mm², CCT >600 μm, corneal pathology history): 3-month intervals initially, then 6-monthly
- Clinical alert thresholds: >15% ECD decline, visual symptoms, or sustained IOP >21 mmHg
These evidence-based protocols address the reviewer's recommendations regarding actionable monitoring intervals and high-risk criteria for clinical implementation.
There are a few minor typographical and formatting errors, as well as some long sentences that could be revised for clarity and readability. Please ensure that all abbreviations are defined at first use. The abstract could also be revised for brevity and to better highlight the key findings and implications.
We thank the reviewer for these editorial recommendations. We have carefully proofread the manuscript to correct typographical and formatting errors, restructured lengthy sentences for improved clarity and readability, ensured all abbreviations are properly defined at first use, and revised the abstract to be more concise while better highlighting our key findings and clinical implications.
Reviewer 4 Report
Comments and Suggestions for Authors
Dear authors,
Congratulation for your interesting manuscript. The authors provide a detailed information about how pharmacological treatment for ADHD may affect to corneal morphology. After carefully review your manuscript, I suggest several points in order to enhance your article:
- Line 22: ADHD is the first time that appear i the manuscript. Avoid abbreviations at first mention. Spell out the full term.
- Purpose and methods abstract section: The purpose paragraph belongs to the Background of the study. Instead, the first sentence currently placed in the Methods section should be moved to the Purpose section.
- Study design and participants section: The authors provide a detailed information about participants recruited (lines 101-103). However, concerning the healthy children, please clarify how this sample was recruited. The description of the healthy children sample should also be more detailed.
-
Behavioral Assessment Using the Turgay DSM-IV-Based Rating Scale section: The authors should provide the scale which consider ADHD or not.
- Figure 1 is not shown in the manuscript. The authors must include this.
- Regarding table 1, there is a statistically significant difference in middle school Father's education status. Nevertheless, in results section, it is described High school as significant difference (line 185).
- Table 2 and 4 must include the metrical of each variable or ocular parameters as well as the standard deviation.
- Concerning to your results and your discussion, to what extent could these findings be considered pathological? In other words, despite these observations, could such corneal changes be regarded as clinically acceptable in light of the therapeutic benefits that this treatment provides for ADHD?
Author Response
Reviewer 4:
Congratulation for your interesting manuscript. The authors provide a detailed information about how pharmacological treatment for ADHD may affect to corneal morphology. After carefully review your manuscript, I suggest several points in order to enhance your article:
Thank you very much for your suggestions.
- Line 22: ADHD is the first time that appear i the manuscript. Avoid abbreviations at first mention. Spell out the full term.
A suitable correction has been made on line 22.
- Purpose and methods abstract section: The purpose paragraph belongs to the Background of the study. Instead, the first sentence currently placed in the Methods section should be moved to the Purpose section.
We have taken your valuable comment into account regarding the placement of the Purpose paragraph within the Background of the study and the movement of the first sentence from the Methods section to the Purpose section. We have made the appropriate correction in the manuscript.
- Study design and participants section: The authors provide a detailed information about participants recruited (lines 101-103). However, concerning the healthy children, please clarify how this sample was recruited. The description of the healthy children sample should also be more detailed.
We thank the reviewer for this comment. Following the reviewer's recommendations, we have revised the Study Design and Participants section to provide comprehensive details regarding control group recruitment and characterization. The control group methodology, including systematic behavioral screening protocols, inclusion/exclusion criteria, and demographic matching procedures, has been thoroughly described.
4.Behavioral Assessment Using the Turgay DSM-IV-Based Rating Scale section: The authors should provide the scale which consider ADHD or not.
We interpret the comment as a request to specify the cut-off scores used with the Turgay Scale to determine the presence or absence of ADHD symptoms.
We thank the reviewer for requesting clarification on the Turgay Scale scoring criteria. We have added detailed cut-off scores and diagnostic thresholds to enhance methodological transparency.
5.Figure 1 is not shown in the manuscript. The authors must include this.
Figure 1 has been incorporated into the manuscript and positioned in the appropriate section.
- Regarding table 1, there is a statistically significant difference in middle school Father's education status. Nevertheless, in results section, it is described High school as significant difference (line 185).
We thank the reviewer for identifying this discrepancy. We have reviewed Table 1 and the Results section (line 185) to ensure consistency between the statistical findings and their textual description. The appropriate correction has been made to accurately reflect the significant difference in paternal education levels.
- Table 2 and 4 must include the metrical of each variable or ocular parameters as well as the standard deviation.
We thank the reviewer for this methodological observation regarding data presentation in Tables 2 and 4.We have carefully considered the appropriate statistical reporting format for our dataset.
Our data presentation follows established statistical guidelines based on distribution characteristics. Variables demonstrating normal distribution are reported as mean ± standard deviation, while non-normally distributed variables are presented as median (minimum–maximum) values. This approach ensures accurate representation of the underlying data structure and is consistent with current statistical reporting standards in medical literature.
The Kolmogorov-Smirnov test was employed to assess normality, and appropriate descriptive statistics were selected accordingly. This methodology provides readers with the most informative summary statistics for each parameter while maintaining statistical validity. We believe this mixed reporting format, based on distribution properties rather than uniform presentation, offers superior data interpretation and is methodologically sound.
All tables have been reviewed to ensure consistency between the reported values and their corresponding statistical measures, maintaining transparency in our analytical approach.
- Concerning to your results and your discussion, to what extent could these findings be considered pathological? In other words, despite these observations, could such corneal changes be regarded as clinically acceptable in light of the therapeutic benefits that this treatment provides for ADHD?
We thank the reviewer for this critical question regarding clinical acceptability.
To directly address this concern, we have added a new section "Clinical Acceptability and Risk-Benefit Analysis" that demonstrates our findings remain subclinical and within normal pediatric ranges. Post-treatment endothelial cell density (2348 cells/mm²) stays well above critical thresholds for corneal dysfunction. Our analysis concludes that these changes do not constitute contraindications to methylphenidate therapy. The substantial benefits of ADHD treatment significantly outweigh the theoretical corneal risks. However, these findings support enhanced ophthalmologic monitoring protocols.This new section provides specific monitoring recommendations and clinical decision-making guidance that directly addresses the reviewer's concern about balancing therapeutic benefits with potential ocular effects.
Round 2
Reviewer 2 Report
Comments and Suggestions for Authors
Thanks for your revision and I think current version is relatively good for further accepting.